# Magnesite as a Sorbent in Fluid Combustion Conditions—Role of Magnesium in SO₂ Sorption Process

**Elżbieta Hycnar** [1,*], **Magdalena Sęk** [1] **and Tadeusz Ratajczak** [2]

1  Department of Mineralogy, Petrography and Geochemistry, University of Science and Technology in Cracow, 30-059 Cracow, Poland

2  Mineral and Energy Economy Research Institute of Polish Academy of Science in Cracow, 31-261 Cracow, Poland

*  Correspondence: hycnar@agh.edu.pl; Tel.: +48-12-617-41-42

**Abstract:** This article presents the results of research on magnesites from the Polish deposits of Szklary, Wiry and Braszowice as SO₂ sorbents under the conditions of fluidized bed combustion technology. In practice, magnesites are not used as SO₂ sorbents, and the role of magnesium in the desulfurization process under the conditions of fluidized bed combustion technology is evaluated differently among researchers. The literature data question the participation of magnesium in the process of SO₂ capture from flue gas and prove its high reactivity. Similarly, previous studies referred to the problem of the stability of magnesium-containing desulfurization products under high temperature conditions. This paper analyzes the SO₂ binding process and determines the parameters of the sorbent responsible for the efficiency of magnesite sorption. It was shown that MgO, formed as a result of thermal dissociation of magnesite, actively participates in the SO₂ binding reaction to form magnesium sulfate phases (MgSO₄ and CaMg₂(SO₄)₃) stable in the temperature conditions of fluidized bed boilers. The problem of differentiated reactivity of magnesium-containing sorbents should be associated with the porosity of the sorbents. If the secondary surface of the sorbent is developed based on micropores and smaller mesopores (below 0.1 μm), the sorbent will be characterized by low sorption activity. It was shown that the SO₂ binding process is then limited only to the outer part of the sorbent grains. This results in the formation of a massive, SO₂-impermeable desulfurization-product layer on the sorbent grain surface. In real conditions, where the reactions of CaCO₃ thermal dissociation and SO₂ sorption occur almost simultaneously, the inside of the sorbent grains may remain undissociated. The results of experimental research allowed us to trace the dynamics of the SO₂ binding process in relation to real conditions prevailing in fluidized bed boilers.

**Keywords:** SO₂ sorbents; magnesites; Ca and Mg sulfates; desulfurization products; fluidized bed combustion technology





## 1. Introduction

Carbonate rocks are widely used in the power industry, where they function as sorbents in flue gas desulfurization technologies. Limestones are mainly used, both in wet and dry or semi-dry desulfurization technologies, as well as in fluidized bed combustion technology. This is related to both the high availability of these rocks and, consequently, lower purchase costs, as well as high desulfurization efficiency. It is also advantageous to be able to use a desulfurization product which takes the form of gypsum (CaSO₄·2H₂O), also called desulfogypsum, a valuable raw material for a wide range of applications.

In contrast to limestone, for carbonate rocks containing magnesium, fewer desulfurization methods have been developed that can be used on an industrial scale. Magnesites, and the product of their calcination (MgO), are used as sorbents in the wet method of desulfurization. This method, like the wet limestone method, is waste-free. The product of the desulfurization reaction in this case is magnesium sulfate heptahydrate (MgSO₄·7H₂O),

used in agriculture and gardening as a fertilizer, and magnesium chloride hexahydrate ($MgCl_2 \cdot 6H_2O$), used as road salt [1].

A variant of the wet magnesium method with regenerated MgO is also mentioned in the literature. In this case, the product of the desulfurization reaction is magnesium sulfite ($MgSO_3$). The MgO recovery is carried out by heating $MgSO_3$ at a temperature of 900–1000 °C. Regenerative methods of desulfurization are more expensive. However, in the absence of the capabilities of using desulfurization products, they are an attractive alternative, giving the possibility of repeated use of the sorbent, which can significantly reduce the costs of gas desulfurization. It should be mentioned that if sulfates are formed as a product of desulfurization, the possibility of regenerating the sorbent will be problematic and may even become impossible [2].

Dolomites and magnesites, as well as limestones with increased magnesium content, apart from wet desulfurization methods, are not used in practice as $SO_2$ sorbents. The possibilities of using this type of rock remain in the sphere of science research. The research results indicate that carbonate rocks with a high magnesium content can be highly effective $SO_2$ sorbents in fluidized bed furnaces. It should be noted, however, that the results of research by other authors differ in this case. Some of them completely doubt the use of these rocks as $SO_2$ sorbents in fluidized bed combustion technology, proving that the magnesium element (MgO) does not participate in $SO_2$ binding and should be treated as non-reactive ballast [3–5]. Many studies show the participation of magnesium in the $SO_2$ bonding reaction, but the thermodynamic durability of desulfurization products containing magnesium in the structure, under temperature conditions typical for fluidized bed furnaces, is questioned. The literature also indicates a lower efficiency of these rocks in the $SO_2$ binding process [6–8]. There are also papers in which the results of the research confirm the effective participation of MgO in $SO_2$ binding, based on the presence of both calcium and magnesium ($CaMg_2(SO_4)_3$) and magnesium ($MgSO_4$) sulfates among the desulfurization products. The research also confirmed the durability of the main product of desulfurization, i.e., $CaMg_2(SO_4)_3$, in temperature conditions prevailing in fluidized bed furnaces (up to 850 °C) [9]. The effectiveness of dolomite desulfurization in fluidized bed furnaces, as compared to lime sorbents, has been shown in tests and is similar or even higher [9–15]. In this case, the structural and textural parameters of the sorbent, produced during the thermal dissociation process, have a decisive impact. This is related to the two-stage process of thermal dissociation of dolomite with the formation of a short-lived calcite phase [9]. The high efficiency of dolomites in the process of $SO_2$ sorption in the conditions of fluidized bed furnaces, demonstrated by tests, became the basis for undertaking research on the use of magnesites.

The aim of the research presented herein was to demonstrate the active participation of magnesium in the $SO_2$ binding process under temperature conditions characteristic of fluidized furnaces. Magnesites were used in the research. The research material was selected in such a way as to:

— eliminate the significant role of calcium in the $SO_2$ capture process. This made it possible to determine the thermal stability of the newly formed sulfate phases containing magnesium.

— show that the high content of magnesium-containing carbonate phases does not guarantee high desulfurization efficiency. An important role in this case is played by the secondary porosity of the sorbent, determined by the presence of pores with diameters bordering on mesopores and macropores.

Much space is devoted to the mineralogical and petrographic characteristics of the studied magnesites, as we intended to show that both the mineral composition and the structural and textural character of the rock may have a significant impact on the desulfurization efficiency and may be responsible for the reduced sorption capacity of the sorbent.

## 2. Materials and Methods

The materials for the research were magnesites from Polish deposits located in Lower Silesia–Braszowice, Szklary, Wiry (Table 1 and Figure 1). The magnesite deposits belong to a geological unit called the Fore-Sudetic Block. The magnesite occurs here in serpentinite rocks, forming the Braszowice–Grochowa and Szklary massifs. The serpentinite massifs were formed as a transformation result of the ultrabasic igneous intrusions composed of peridotites and dunnites under the influence of hydrothermal solutions. The tropical climate prevailing in the Tertiary period caused intense chemical weathering of the serpentines, which resulted in, among other products, the formation of magnesite. In these deposits, magnesite takes the form of sockets and veins of varying thickness, which, branching, become dispersed, turning into mesh-like fine veins crossing the serpentine massif [16]. In such an occurrence, magnesite is white, white with a yellow rim, or yellow and brown. The varied color of magnesite is the result of a changeable chemical composition and depends on the degree of weathering of the serpentinite. The different colors depend on the iron content. White magnesite, characterized by high purity, occurs in serpentinite not affected by weathering processes.

**Table 1.** List of magnesites used for the evaluation of sorption properties in relation to $SO_2$ in the conditions of fluid combustion technology.

| Sample Number | 1 | 2 | 3 |
| --- | --- | --- | --- |
| Origin | Magnesite from the Szklary deposit | Magnesite from the Wiry deposit | Magnesite from the Braszowice deposit |
| | from the heap | from the heap | from the deposit |

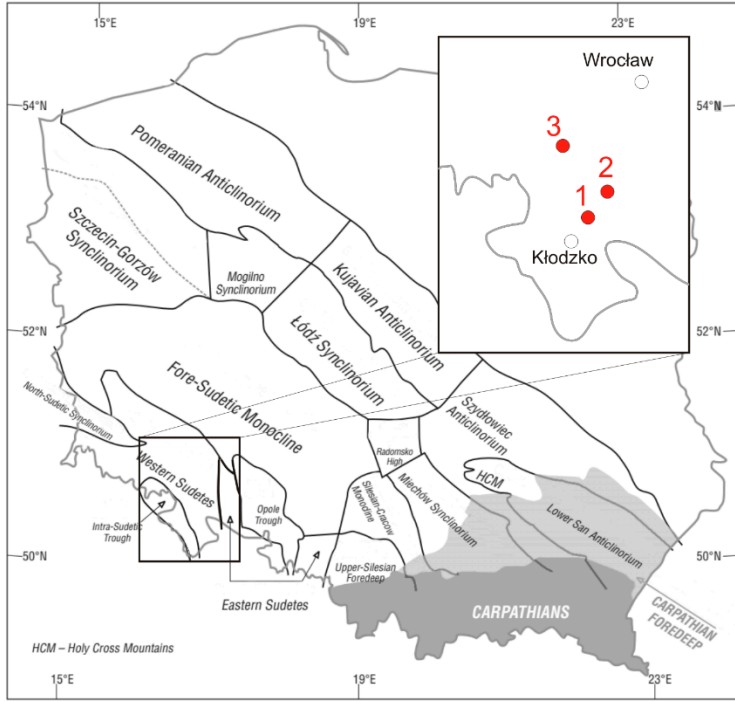

**Figure 1.** Locations of magnesite deposits from which materials were collected for this research, against the background of the main tectonic units on the sub-Cenozoic surface of Poland [17]. 1—Szklary deposit; 2—Wiry deposit; 3—Braszowice deposit.

Among the mentioned deposits, only the Braszowice deposit is currently exploited. In this case, the samples were taken from the operational mining walls of the "Konstanty" opencast mine. The remaining samples were obtained from the heaps located in the areas

of closed mines. For comparison purposes, high-quality lime industrial sorbent was used in parallel studies and was treated as a reference material.

The objective of this research was to determine the role of magnesium in the process of flue gas desulfurization under the conditions of fluidized bed furnaces. Particular attention was paid to the mineralogical and petrographic characteristics of the magnesites used as $SO_2$ sorbents and sulfation reaction products. An attempt was made to:

— depict the mechanism of $SO_2$ sorption by magnesite.
— determine the parameters of the sorbent responsible for the effectiveness of the $SO_2$ binding process.

The results of this research were compared with the results of our earlier research regarding the possibility of using dolomites as $SO_2$ sorbents in fluidized bed combustion conditions [9]. This comparison was possible thanks to the use of the same measurement conditions during the sulfation experiment and the research methodology. Comparing research results with the works of other authors is a great challenge and is often not possible due to the varied conditions of conducting the experiments, incomplete mineralogical characteristics of both the research material (which is of great importance when conducting research with the use of natural raw materials) and sulfation products. The authors' own experience in the study of this type of sorbents indicates that the use of mineralogical and petrographic research methods will allow, in addition to determining the magnesium content, obtaining a more complete picture of the $SO_2$ sorption mechanism and the impact of sorbent parameters other than chemical composition on the $SO_2$ binding process.

To assess the sorption properties of magnesites, the research methodology developed for the purpose of testing dolomites, presented in [9], was used.

The sulfation experiment was carried out based on the guidelines developed by Ahlstrom Development Laboratory [18]. This method is based on determining two indicators: the reactivity (RI index) and absolute sorption (CI index). The reactivity index determines the ratio of the calcium content (in the case of magnesite studies, magnesium was additionally included) in the sample to the amount of sulfur after the sorption process [Ca + Mg/S moles]. The absolute sorption index, in turn, determines the amount of sulfur sorbed by 1000 g of the sorbent [g S/1000 g of the sorbent]. The $SO_2$ sorption studies were carried out using a material with a particle size of 0.125–0.250 mm. The sulfation experiment was carried out based on a gas-tight retort furnace acting as a fixed bed. The sulfation experiment was carried out in two stages in accordance with the guidelines:

1.  Samples were subjected to a decarbonization process at 850 °C for 30 min prior to sulfation. A sample of the sorbent (150 mg) was placed inside the combustion chamber, on a perforated ceramic plate, in such a way that the individual grains of the sorbent were not in contact with each other. In this way, free access of gases to individual sorbent grains was ensured during the experiment. Synthetic air containing 80% $N_2$ and 20% $O_2$ was passed through the samples.
2.  Then, a gas containing 1780 ppm of $SO_2$, 3% of $O_2$, 16% of $CO_2$ and $N_2$ was passed through the samples at a speed of 950 mL/second for another 30 min.

In the next stage, the content of absorbed sulfur was determined with an elemental analysis apparatus for carbon, hydrogen, nitrogen and sulfur (Series 628, LECO). The results of the research became the basis for calculating the values of *RI* and *CI* indicators according to the formulas:

$$RI = \frac{\frac{M_S}{100}\left(\frac{x_{Ca}}{M_{Ca}} + \frac{x_{Mg}}{M_{Mg}}\right)\left(1 - \frac{M_{CO_2}}{M_C} \cdot \frac{x_{C_p}}{100} - \frac{M_{SO_3}}{M_S} \cdot \frac{x_{S_p}}{100}\right)}{\frac{x_{S_p} - x_{S_b}}{100} + \frac{M_{CO_2}}{M_C} \cdot \left(\frac{x_{C_p} \cdot x_{S_b} - x_{C_b} \cdot x_{S_p}}{10{,}000}\right)} \tag{1}$$

$$CI = \frac{1000 \cdot \left[\frac{x_{S_p} - x_{S_b}}{100} + \frac{M_{CO_2}}{M_C} \cdot \left(\frac{x_{C_p} \cdot x_{S_b} - x_{C_b} \cdot x_{S_p}}{10{,}000}\right)\right]}{1 - \frac{M_{CO_2}}{M_C} \cdot \frac{x_{C_p}}{100} - \frac{M_{SO_3}}{M_S} \cdot \frac{x_{S_p}}{100}} \tag{2}$$

where $x_{Ca}$, $x_{Cp}$, $x_{Sp}$, $x_{Cb}$, $x_{Sb}$—percentages of calcium in the sorbent, carbon in the sorbent after the sulfating process, sulfur after the sulfating process, carbon before the sulfating process and sulfur before the sulfating process, respectively [%]; $M_S$, $M_{Ca}$, $M_{Mg}$, $M_C$, $M_{CO2}$, $M_{SO3}$—molar masses of sulfur, calcium, magnesium, carbon, carbon dioxide and sulfur trioxide, respectively [kg/kmol].

To evaluate the sorption capacity, the five-level scale proposed by Ahlstrom Development Laboratory was used (Table 2).

**Table 2.** Reference values of the reactivity (RI) [Ca moles/S moles] and the absolute sorption (CI) [g S/1000 g of the sorbent] [18].

| Sorption Capacity of the Sorbent | RI [kmol Ca+Mg/kmol S] | CI [g S/1 kg of Sorbent] |
|---|---|---|
| Excellent | <2.5 | >120 |
| Very good | 2.5–3.0 | 100–120 |
| Good | 3.0–4.0 | 80–100 |
| Sufficient | 4.0–5.0 | 60–80 |
| Low quality | >5.0 | <60 |

Note: in the above test, the measure of the reactivity of the tested raw material is the value of the RI index, which has been modified to consider the participation of magnesium in the sorption process.

To characterize the parameters affecting the $SO_2$ binding process, the experimental studies were supplemented with:

- Microscopic observations in polarized transmitted light were made using an Olympus BX-41 (Olympus, Tokyo, Japan) polarizing microscope with an Olympus SC180 camera.
- Tests of the phase composition and structural and textural features of natural samples after the decarbonation and sulfation process using X-ray diffraction using the DSH method (MiniFlex 600, Rigaku, Tokyo, Japan) and scanning microscopy (Quanta 200 FEG, FEI, Hillsboro, OR, USA).
- Research of the distribution of Ca, Mg and S within sorbent grains using an electron probe microanalyzer (EPMA) (JEOL Super Probe 8230, Peabody, MA, USA).
- Analysis of the main chemical components was performed using instrumental methods, i.e., atomic emission absorption spectroscopy (ICP–OES Plasma 40, PerkinElmer, Waltham, MA, USA) and complexometric titration (determination of $MgCO_3$ and $CaCO_3$).
- The temperature and course of thermal dissociation of magnesites (STA 449 F3 Jupiter + QMS 403C Aelos, Netzsch, Selb, Germany) were determined with differential derivatographic analysis (DTA) and thermogravimetry (TG). In addition, thermal analyses (TGA/DSC 3+, Mettler Toledo, Greifensee, Switzerland) were performed for the products of the sulfation experiment in order to determine their thermal stability in high-temperature conditions. All measurements were made in an air atmosphere.
- A porous texture analysis was performed using a mercury porosimeter. The following porous texture parameters were determined:

1. The coefficient of effective porosity, i.e., the ratio of pore volume to the total volume of the sample [19–22]:

$$\phi = \frac{V_{tot}}{V_b} \cdot 100\% = \frac{V_b - V_s}{V_b} \cdot 100\% = \left(1 - \frac{\rho_b}{\rho_s}\right) \cdot 100\% \tag{3}$$

where $\phi$—coefficient of effective porosity (%); $V_{tot}$—total volume of mercury in the pores (mL); $V_b$—external volume (mL); $V_s$—skeleton volume (mL); $\rho_b$—bulk density (g/mL); $\rho_s$—skeletal density (g/mL).

2. The specific surface area of porous space, i.e., the pore area in relation to the sample unit mass. This parameter characterizes the flow resistance of reservoir media in the porous medium. The specific surface area, assuming the reversibility of the injection process, is determined based on the obtained pore volume according to the following equation [19,22]:

$$A = -\frac{1}{\gamma \cos\theta} \int_0^V PdV \tag{4}$$

where A—total surface area of porous space (m$^2$/g); dV—partial pore volume corresponding to the given capillary pressure (m$^3$); P—capillary pressure (psi); $\gamma$—surface tension of mercury (dyna/cm); $\theta$—contact angle ($°$).

The study of the specific surface area and porosity was also carried out using the low-temperature nitrogen sorption method. An ASAP 2020 device for precise sorption measurements (Micromeritics, Norcross, GA, USA) was used for the analysis. Immediately before measurement, the samples were dried at 105 $°$C for 1 h, and then annealed under vacuum at 150 $°$C for 12 h. The specific surface area (S$_{BET}$) was determined based on low-temperature nitrogen adsorption isotherms at $-196$ $°$C according to the Brunauer–Emmet–Teller (BET) method [23]. The total pore volume ($V_{tot}$) was also calculated for the relative pressure p/p0 = 0.99. In addition, we calculated:

1. the micropore volume ($V_{mik}{}^{DR}$), for pores with a width of less than 2 nm, according to the Dubinin–Radushkevich method, and their share in the total pore volume $V_{mik}{}^{DR}/V_{tot}{}^{0.99}$;
2. the mesopore volume ($V_{mez}{}^{BJH}$), for pores with a diameter above 2 nm and below 50 nm, according to the Barrett–Joyner–Halenda (BJH) method, and their share in the total pore volume ($V_{mez}{}^{BJH}/V_{tot}{}^{0.99}$);
3. the macropore volume ($V_{mak}$), for pores with a diameter above 50 nm, calculated by subtracting the volume of micro- and mesopores from the total pore volume: $V_{mak} = V_{tot}{}^{0.99} - (V_{mik}{}^{DR} + V_{mez}{}^{BJH})$, and their share in the total pore volume $V_{mak}/V_{tot}{}^{0.99}$.

The analyses were performed in accordance with standards: ISO 9277:2010 [24], ISO 15901-2:2006 [25] and ISO 15901-3:2007 [26].

## 3. Results and Discussion

### 3.1. Mineralogical and Petrographical Characters and Chemical Composition of Magnesites

The studied magnesites are generally microcrystalline rocks. They are made of magnesite (MgCO$_3$) crystals with a size of 2–5 $\mu$m (Figures 2 and 3).

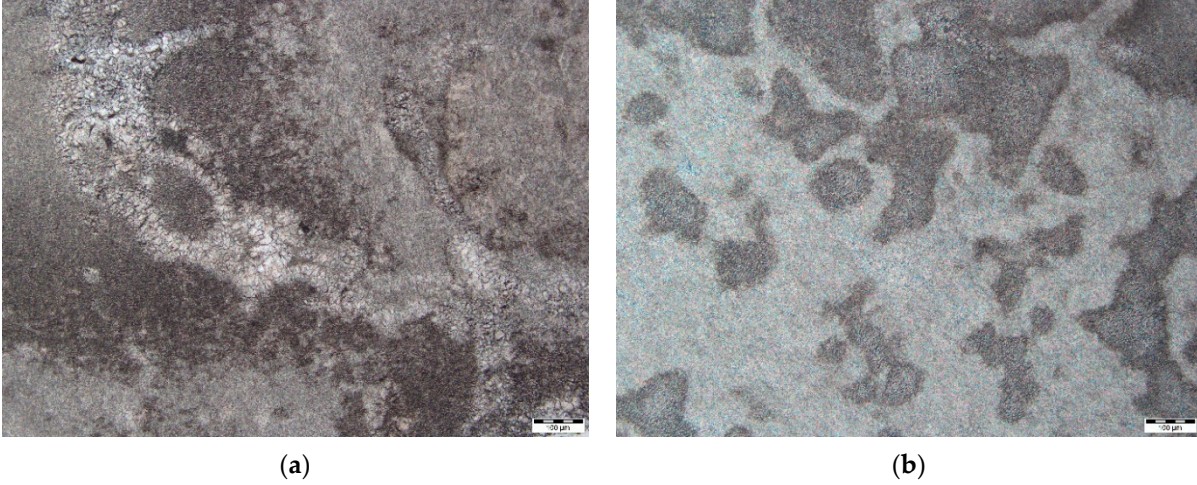

(**a**)　　　　　　　　　　　　　　　　　　　　　　　(**b**)

**Figure 2.** Magnesite from the Wiry deposit. (**a**,**b**) Filled with magnesite cement and dark areas pigmented with iron compounds.

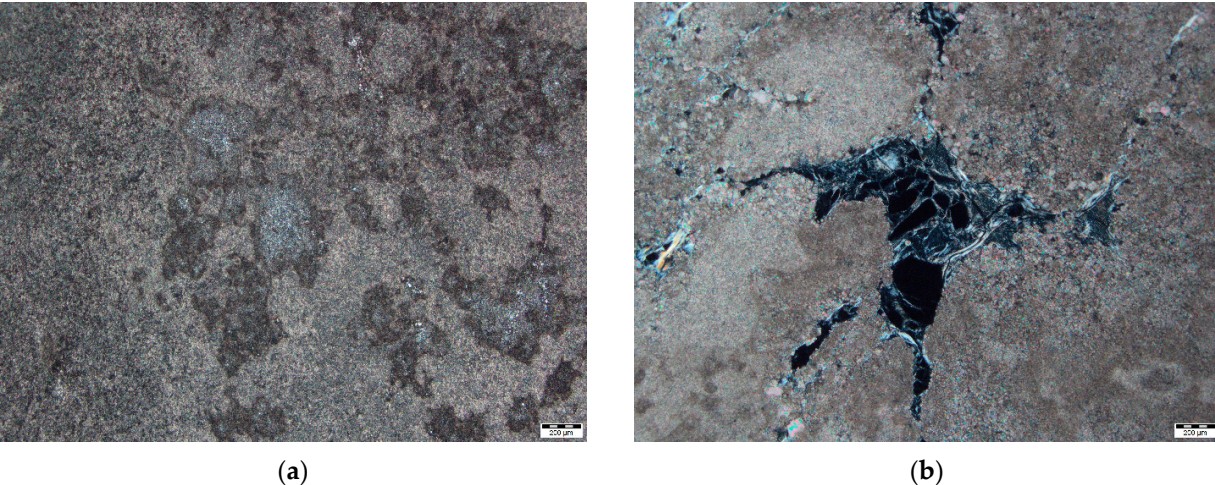

(**a**)                                                                                    (**b**)

**Figure 3.** Magnesite from the Szklary deposit. (**a**) Rock pores filled with magnesite cement and dark-colored areas pigmented with iron compounds. (**b**) Karst pores partially filled with minerals from the serpentine group, sparite crystals of magnesite.

In the magnesites from the Szklary and Wiry deposits, there are numerous cavities and pores, formed as a result of the $MgCO_3$ leaching processes. They were filled with larger (microsparite and sparite) magnesite crystals (Figures 2 and 3). These types of fillings or areas in their immediate vicinity are often pigmented with iron compounds and their texture is clearly porous (Figures 2 and 3).

The magnesite from the Braszowice deposit is characterized by a colomorphic structure. The microcrystalline and cryptocrystalline created $MgCO_3$ forms irregular laminae of varying thickness from 0.05 to 0.5 mm and a complicated course, highlighted by a color change caused by the presence of iron compounds (Figure 4). Within the laminas, a gradational increase in the size of magnesite crystals to several μm is observed. The texture of the magnesite from Braszowice is massive. The pore space between the magnesite crystals is filled with clay minerals; unfilled pores are rare. They can be seen between successive laminates of magnesite. Usually within them, there are minerals from the serpentine group, most often a fibrous clinochrysotile.

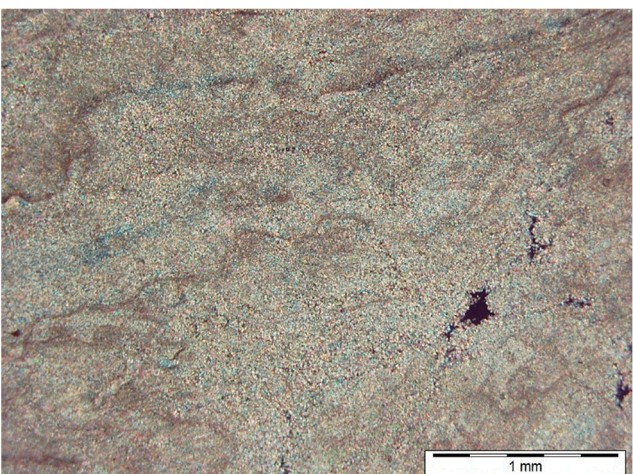

**Figure 4.** Magnesite from the Braszowice deposit, showing irregular lamination and empty pores.

In the studied magnesites, a silification process was observed, manifested by the presence of chalcedony aggregates and microcrystalline quartz within the rock pores.

The content of non-carbonate minerals in the rocks was determined by means of quantitative microscopic analysis. In the magnesites from Wiry, their presence was estimated at

about 5% of the rock volume. These are mainly the above-mentioned minerals from the silica group. In the magnesites from Szklary and Braszowice, the share of non-carbonate minerals is higher, on the order of 10 and 12% by volume. In addition to quartz and chalcedony, minerals from the serpentine group have been identified, occurring within rock pores that are part of the karst system (Figure 2b). It is believed that they were transported there via karst solutions. In the magnesites from Braszowice, the pore space of the rock is additionally filled with clay minerals.

In the mineral composition of the examined rocks, the presence of ferruginous type magnesite was additionally confirmed by diffraction tests (Table 3 and Figure 5). In the samples of rocks from the Szklary and Wiry deposits, the addition of dolomite (up to 5% of the rock volume) and trace amounts of calcite were identified, mainly in the case of magnesite from Szklary (up to 3% of the rock volume). Clay minerals of the illite/montmorillonite type were identified in the composition of the magnesites from Szklary and Braszowice. In the magnesite from Szklary, sepiolite, which is a typical product of serpentinite conversion under the influence of weathering processes, and talc, formed as a hydrothermal olivine and pyroxene transformation result, also marked their presence.

**Table 3.** Phase composition of the magnesites identified by diffraction analysis.

| Phase Components | Magnesites | | |
| --- | --- | --- | --- |
| | From the Szklary Deposit | From the Wiry Deposit | From the Braszowice Deposit |
| Magnesite | + | + | + |
| Iron magnesite | + | + | + |
| Dolomite | + | + | − |
| Calcite | + | + | − |
| Quartz | + | + | + |
| Illite/montmorillonite | + | − | + |
| Sepiolite | + | − | − |
| Talc | + | − | − |

Explanations: (+) means the presence of a mineral phase in the composition of the sample; (−) means the presence below the detection limit, i.e., 3% of the rock volume.

The chemical composition of the investigated magnesites reflects their phase composition. It shows a slight quantitative variation in terms of the key components for the course of $SO_2$ sorption content, such as $CaCO_3$ and $MgCO_3$ (Table 4).

**Table 4.** Chemical composition of the investigated magnesites [% wt.].

| Component | Magnesites | | | Industrial Sorbent |
| --- | --- | --- | --- | --- |
| | From the Szklary Deposit | From the Wiry Deposit | From the Braszowice Deposit | |
| $SiO_2$ | 6.61 | 5.42 | 6.89 | 0.53 |
| $Al_2O_3$ | 1.42 | 0.22 | 3.14 | 0.28 |
| $Fe_2O_3$ | 3.13 | 0.52 | 3.26 | 0.24 |
| CaO | 2.93 | 1.72 | 0.73 | 54.58 |
| MgO | 43.77 | 41.25 | 41.72 | 0.33 |
| $Na_2O$ | 0.04 | 0.01 | 0.08 | 0.01 |
| $K_2O$ | 0.03 | 0.01 | 0.03 | 0.01 |
| Ignition loss | 41.49 | 50.47 | 43.85 | 44.021 |
| $CaCO_3$ | 5.23 | 3.07 | 1.30 | 97.41 |
| $MgCO_3$ | 79.01 | 86.29 | 73.67 | 0.69 |
| Sum of carbonate | 84.24 | 89.36 | 82.98 | 98.10 |

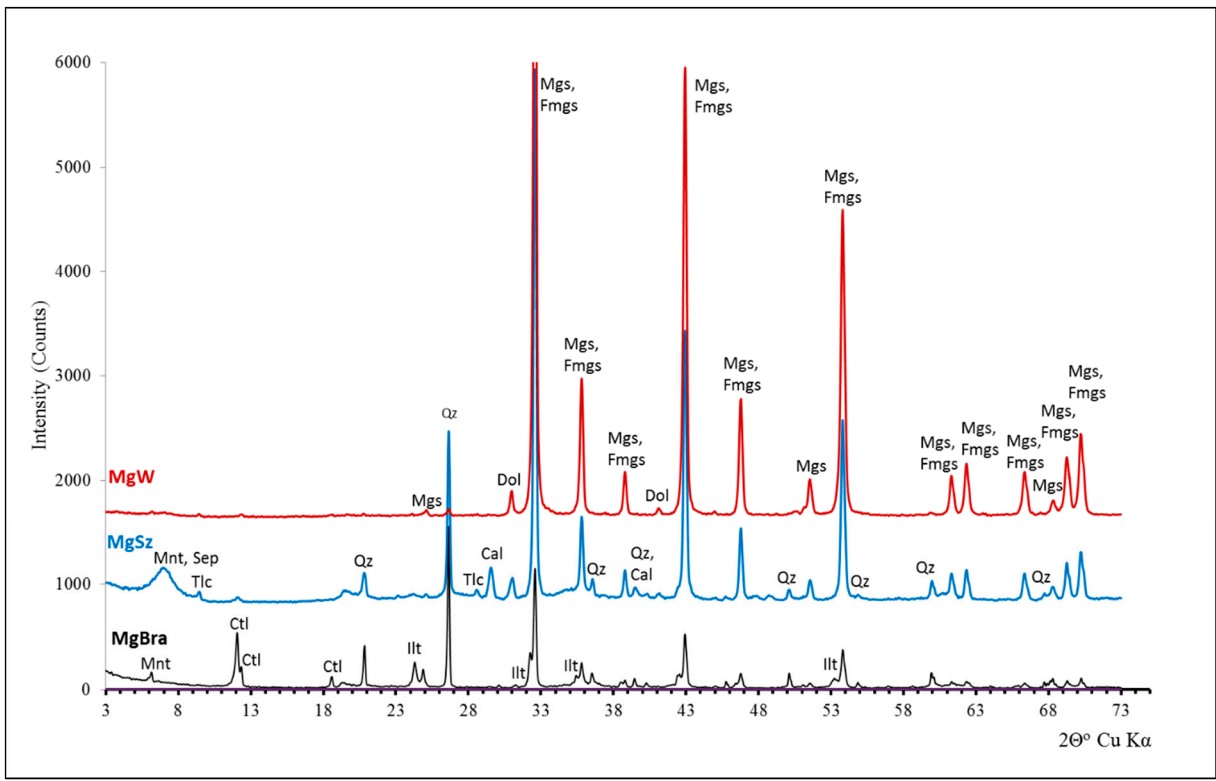

**Figure 5.** Phase compositions of magnesite from the Braszowice deposit (MgBra), Szklary deposit (MgSz) and Wiry deposit (MgW) are shown in the diffraction patterns. Explanations: Mnt—montmorillonite $(Na,Ca)_{0,3}(Al,Mg)_2Si_4O_{10}(OH)_2 \cdot nH_2O)$; Sep—sepiolite $(Mg_4(Si_6O_{15})(OH)_2 \cdot 6H_2O)$; Tlc—talc $(Mg_3Si_4O_{10}(OH)_2)$; Ctl—clinochrysotile $(Mg_3(Si_2O_5)(OH)_4)$; Ilt—illite $(K_{0.65}Al_{2.0}[Al_{0.65}Si_{3.35}O_{10}](OH)_2)$; Qz—quartz $(SiO_2)$; Mgs—magnesite $(MgCO_3)$; Fmgs—ferroan magnesite $((Mg,Fe)CO_3)$; Cal—Calcite $(CaCO_3)$; Dol—dolomite $(CaMg(CO_3)_2)$.

The highest total content of carbonate components is characterized by magnesite from Wiry (89.36% wt). In the case of other magnesites, the contents are lower: 82.98% wt. for magnesite from the Braszowice and 84.2% wt. for magnesite from the Szklary deposit. A small share of calcium carbonate (1.0–5.23% wt.) in the composition of individual samples is noteworthy. The content of $SiO_2$ is at a similar level of 5.42–6.89% wt. Despite the small share of $Al_2O_3$ (0.22–3.14% wt.) in the magnesite composition, the aluminum content should be considered varied. The highest content of $Al_2O_3$ was found in magnesite from Braszowice, which, in combination with the highest content of $SiO_2$, $K_2O$ and $Na_2O$, confirms the highest share of non-carbonate components in the mineral composition of the rock, demonstrated by microscopic examination.

### 3.2. The Sorption Efficiency and MgO Participation in the SO₂ Binding Process

Table 5 shows the average values of reactivity and sorption indicators for individual magnesites and the industrial sorbent. In the case of the studied magnesites, the RI and CI values are within a wide range (RI 2.43–6.1 kmol Ca + Mg/kmol S; CI 122–17 gS/kg of sorbent), which results in a differentiated assessment of sorption properties: from excellent—magnesite from the Wiry deposit, through very good—magnesite from the Szklary deposit, to low quality—magnesite from the Braszowice deposit. Comparing the RI and CI values of the tested magnesites with a high-class industrial sorbent, it should be noted that the magnesite from Wiry, despite a clearly lower content of carbonate components (78.90% by weight) compared to the industrial sorbent (98.10% wt.), shows effective sorption at a comparable level. The magnesite from Szklary has a slightly lower sorption efficiency compared to the magnesite from Wiry, which may be caused by a lower share of carbonate components in the chemical composition of this magnesite (69.59% wt.).

The RI and CI values obtained for the magnesites from Wiry and Szklary indicate the effective participation of magnesium in the $SO_2$ binding process. Why does the magnesite from Braszowice, despite the higher content of carbonate components compared to the magnesite from Wiry (72.88% wt.), show an extremely low sorption efficiency? It should be assumed that the high content of carbonate components is not a guarantee of high efficiency of $SO_2$ sorption in the analyzed measurement conditions.

**Table 5.** Values of the absolute sorption (CI) [gS/1kg sorbent] and reactivity (RI) [kmol Ca/kmol S] indicators of the investigated magnesites and industrial sorbent tested.

| Reactivity Indexes | Magnesites | | | Industrial Sorbent |
|---|---|---|---|---|
| | From the Szklary Deposit | From the Wiry Deposit | From the Braszowice Deposit | |
| RI | 2.79 | 2.43 | 6.10 | 2.35 |
| CI | 99 | 122 | 17 | 130 |
| Assessment of reactivity | Very good | Excellent | Low quality | Excellent |

The results of the X-ray analysis confirm the participation of magnesium in the $SO_2$ binding process. The phase composition of magnesite from Wiry samples after the sulfation process, presented in the diffractograms (Figure 5), shows that the magnesium (MgO) and calcium (CaO) oxides formed at 850 °C (Figure 6A) take an active part in the $SO_2$ binding process to form the sulfate phases: $CaSO_4$, $MgSO_4$ and $CaMg_2(SO_4)_3$ (Figure 6B). The presence of unreacted MgO residues cannot be ruled out, since diffraction lines with values of 2θ = 42.88°, d = 2.1072 Å and 2θ = 62.27°, d = 1.2898 Å are characteristic for both MgO and $CaMg_2(SO_4)_3$ (first line) and $CaSO_4$ (second line).

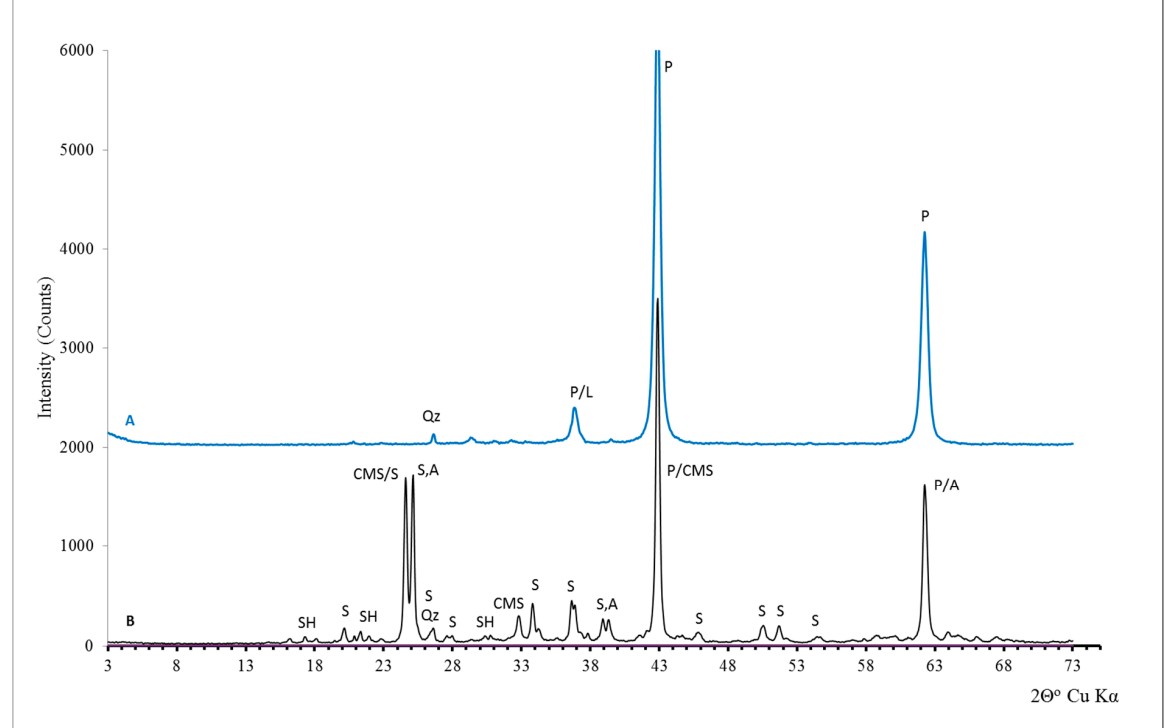

**Figure 6.** Phase compositions of the Wiry deposit: (**A**) after the decarbonization process (–); (**B**) after the sulfation process (–). Explanations: Qz—quartz ($SiO_2$); P—periclase (MgO); L—lime (CaO); SH—magnesium sulfate hydrate ($MgSO_4 \cdot H_2O$); S—magnesium sulfate ($MgSO_4$); CMS—calcium magnesium sulfate ($CaMg_3(SO_4)_4$); A—anhydrite ($CaSO_4$).

To better illustrate the participation of magnesium in the $SO_2$ binding process, distribution maps of Ca, Mg and S were made on the cross-sectional surface of sorbent grains after the sulfation process using an electron probe microanalyzer (EMPA) (Figure 7). The photos show that sulfur accumulates over the entire cross-sectional area of the grain. However, the largest amounts were bound in the near-surface zone and on the outer surface of the grains, as well as along the crack running through the grain. It should be noted that the gap was not covered with desulfurization products during the sulfation process and constituted a diffusion channel for $SO_2$ into the sorbent interior.

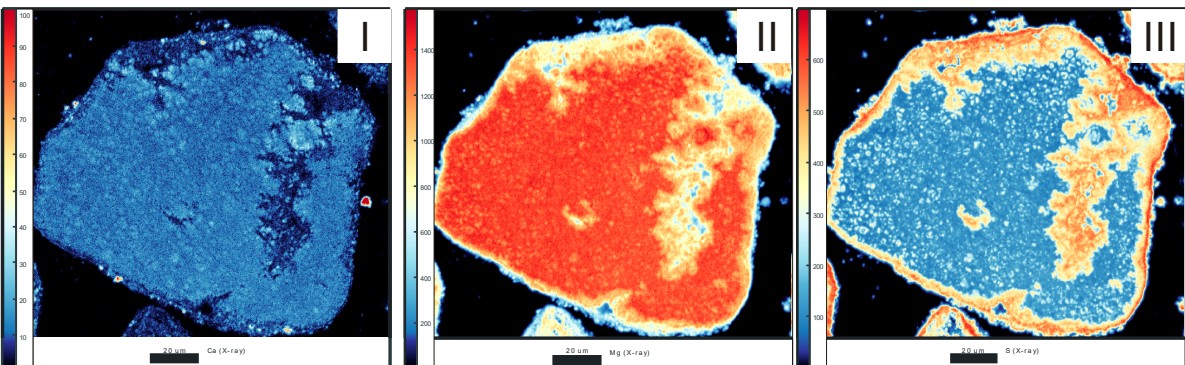

**Figure 7.** Sample of magnesite from Wiry after the sulfation process. Distribution maps of Ca (**I**), Mg (**II**) and S (**III**) in the cross section of the sorbent grain (EMPA).

One of the main arguments given against the use of magnesites as $SO_2$ sorbents in fluidized bed combustion technology is the thermal instability of magnesium sulfates in the temperature conditions characteristic of this technology. In order to investigate the range of thermal stability of the produced sulfates, derivatographic tests were performed. The thermal curves of TG and DTG of magnesite from Wiry after the sulfation process presented in the figure show that the thermal decomposition of $MgSO_4$ begins at a temperature of about 950 °C, and the highest rate of this process occurs at a temperature of about 1050 °C. There is also a faint thermal effect at about 1120 °C, which corresponds to the thermal decomposition of $CaMg_2(SO_4)_3$ (Figure 8). This indicates different properties of sulfates (containing magnesium in their structure), formed in high-temperature conditions in relation to sulfates with an analogous structure, formed from aqueous solutions [27].

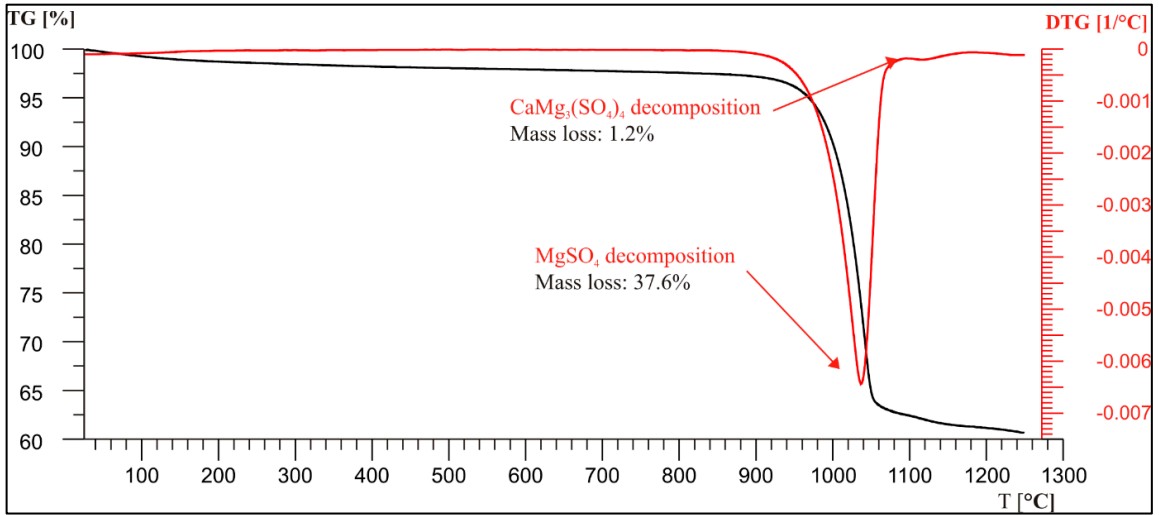

**Figure 8.** TG and DTG thermal curves of magnesite sulfation products from the Wiry deposit.

One of the parameters responsible for the sorption efficiency and the sorbent conversion degree is the thermal dissociation temperature of the carbonate phases involved in the $SO_2$ sorption process [28]. The course of the decarbonation process of magnesites from Wiry and Braszowice was investigated to determine the impact of the mineral composition on it. The thermal effects of the heating of magnesites and the associated mass losses are shown in Table 6 and Figure 9. The thermal decomposition of magnesite from Wiry begins at about 410 °C and is visible as a strong endothermic effect reaching its extreme at 620 °C. At a slightly higher temperature, dolomite decomposes, visible as another endothermic peak from the extreme at 757 °C. It is usually two-stage, but in the case of small amounts of this mineral in the sample, only one endothermic effect is visible during thermal dissociation. At 850 °C, the degree of decarbonation is high and amounts to 50.45% wt. Further heating of the sample does not cause significant changes in mass, because in the temperature range of 850–1000 °C, the mass loss is only 0.02% by wt. Thermal decomposition of magnesite from Braszowice proceeds in a similar way. The process of magnesite decomposition begins at 500 °C and ends at 648 °C with a weight loss of 41.29% wt. At slightly higher temperatures, decomposition of dolomite (738 °C) and calcite (800 °C) is observed. There is also an exothermic reaction associated with the formation of secondary calcite (822.6 °C), which then undergoes thermal dissociation. The decomposition of this type of carbonate phase is accompanied by a negligible mass loss of 1.23% by weight. At 850 °C, the degree of decarbonation is 43.73% wt. Further heating of the sample does not cause significant changes in mass, because in the temperature range of 850–1000 °C, mass loss is only 0.12% by weight. Differences in the mineral composition of the magnesites from Wiry and Braszowice have no significant effect on the course of the carbonation process, apart from slightly lower mass losses during heating, which is related to the lower share of carbonate phases in the magnesite from Braszowice.

**Table 6.** Thermal effects occurring during heating of magnesites from Wiry and Braszowice in the temperature range up to 1000 °C.

| Type of Process | Type of Reaction | Process/Reaction Temperature | | Weight Change |
| | | Beginning | End | |
| | | [°C] | [°C] | [% wag.] |
| --- | --- | --- | --- | --- |
| **Magnesite from Wiry** | | | | |
| Decomposition of $MgCO_3$ | endothermic | 410 | 640 | 49.43 |
| Decomposition of $CaMg(CO_3)_2$ | endothermic | 743 | 770 | 0.71 |
| - | - | 770 | 850 | 0.31 |
| - | - | 850 | 1000 | 0.02 |
| **Magnesite from Braszowice** | | | | |
| Evaporation of surface water | - | 50 | 110 | 1.21 |
| Decomposition of $MgCO_3$ | endothermic | 500 | 648 | 41.29 |
| Decomposition of $CaMg(CO_3)_2$ | endothermic | 700 | 745 | 0.75 |
| Decomposition of $CaCO_3$ | endothermic | 790 | 810 | 0.35 |
| Formation of secondary $CaCO_3$ | exothermic | 810 | 825 | - |
| Decomposition of $CaCO_3$ | endothermic | 825 | 850 | 0.15 |
| - | - | 850 | 1000 | 0.12 |

### 3.3. Porosity of Desulfurization Products and $SO_2$ Sorption Efficiency

The surfaces of decarbonated and sulfated grains, as well as the cross-sections through the sulfated magnesite grains from Wiry and Szklary, presented in Figure 10, make it possible to analyze the $SO_2$ sorption process. The distribution of sulfur inside the sorbent grains is related to the secondary porosity created during the decarbonatization process (Figure 10a,b). This porosity determines the texture of the sulfation products on the sorbent grain surface (Figure 10c,d). The porous texture of the produced sulfates (Figure 10c) ensures the continuity of the $SO_2$ sorption process throughout the sulfation period and is responsible for a higher degree of sorbent grain utilization (Figure 10e). If poorly porous sulfates are produced during the sulfation reaction, as in the case of magnesite from Szklary

(Figure 10d), the SO$_2$ sorption process can be stopped earlier. In such cases, the grains contain a larger unreacted MgO/CaO core (Figure 10e). The sorption efficiency of such a grain and its degree of conversion is lower. Analyzing grain cross-sections, it was observed that in the case of magnesite from Wiry, the sorption process was more effective. Thanks to the porosity visible in Figure 10c, SO$_2$ penetrated the grain almost in its entire volume.

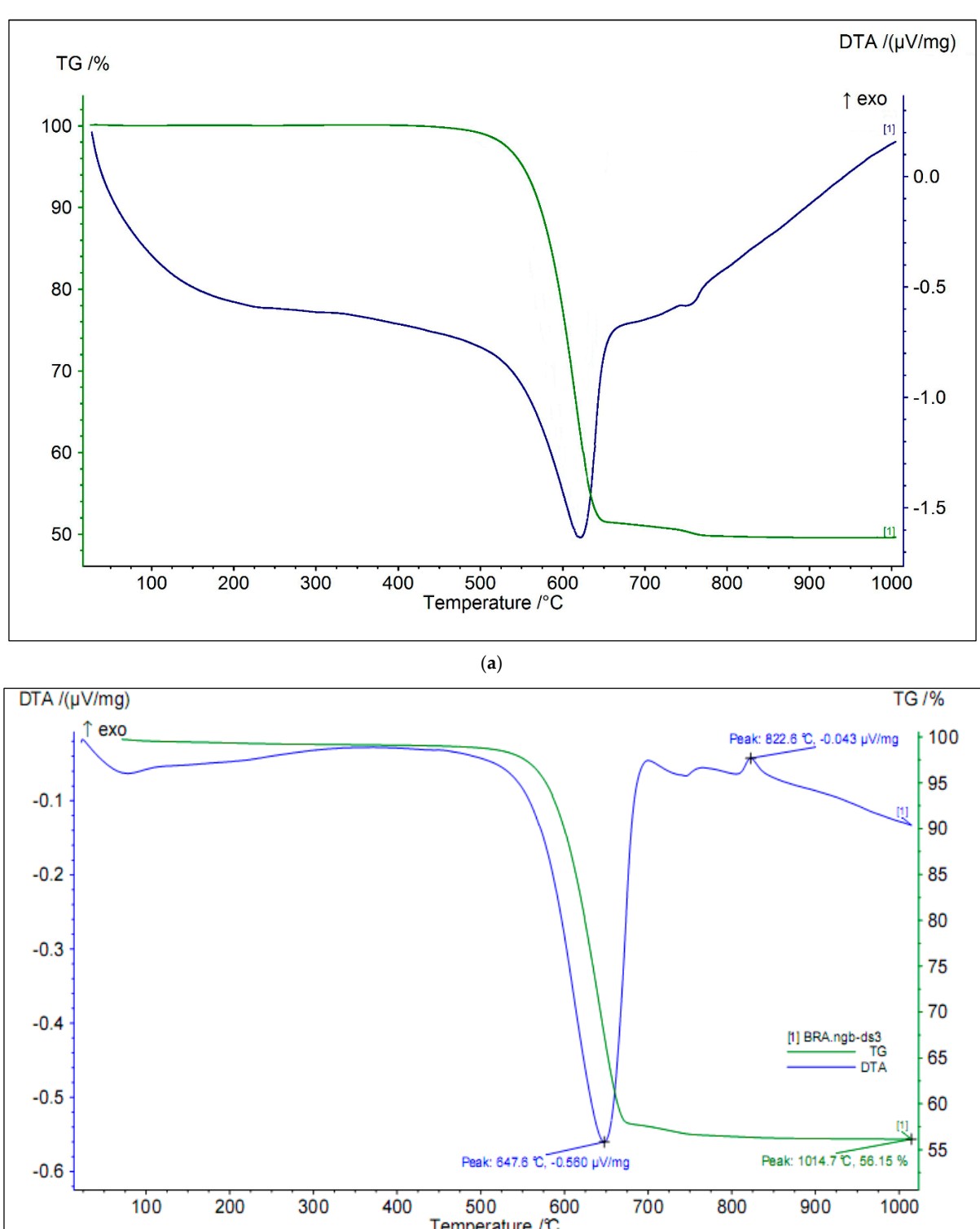

**Figure 9.** Thermal curves (DTA and TG), showing the course of the magnesite thermal dissociation process: (**a**) from the Wiry deposit; (**b**) from the Braszowice deposit.

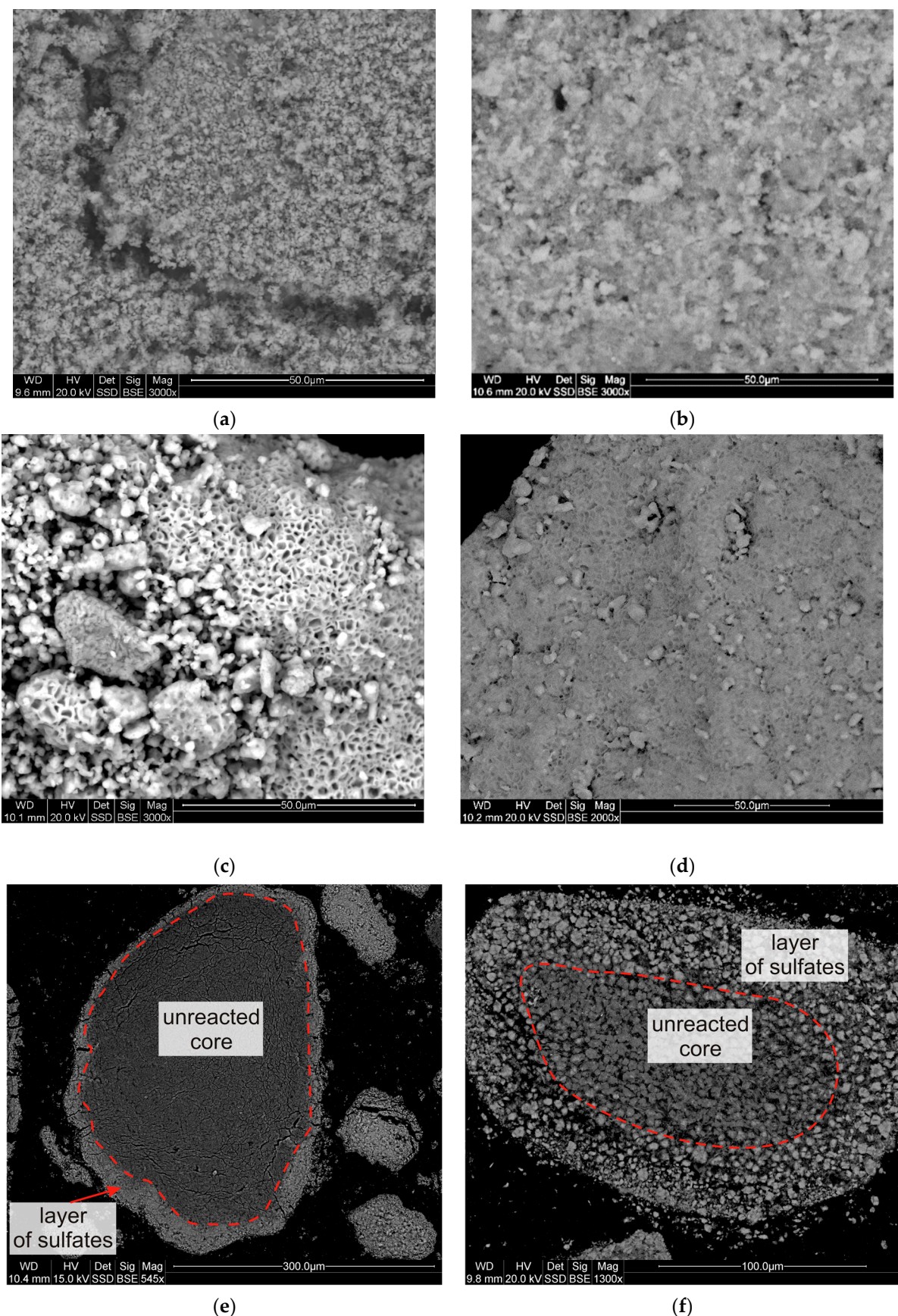

**Figure 10.** Magnesites from Wiry (**a**–**c**) and Szklary (**d**–**f**): (**a**,**b**) samples after the decarbonation process; (**c**–**f**) after the sulfation process; (**c**,**d**) grain surface; (**e**,**f**) cross-section through the grain.

The surface morphology of the magnesite grains from Braszowice, visible in Figure 11a, does not differ in any special way from other magnesites. Despite this, the surface of the grains after the decarbonization process is differently shaped. It has been expanded to a lot greater extent than the magnesites from Wiry and Szklary. The pores formed as a result of thermal dissociation appear to be much smaller (Figure 11b). During the sulfation process, the surface of the grains was covered with massive, non-porous sulfate, which led to stopping the $SO_2$ sorption process in the initial stage and resulted in a low degree of their conversion (Figure 11d).

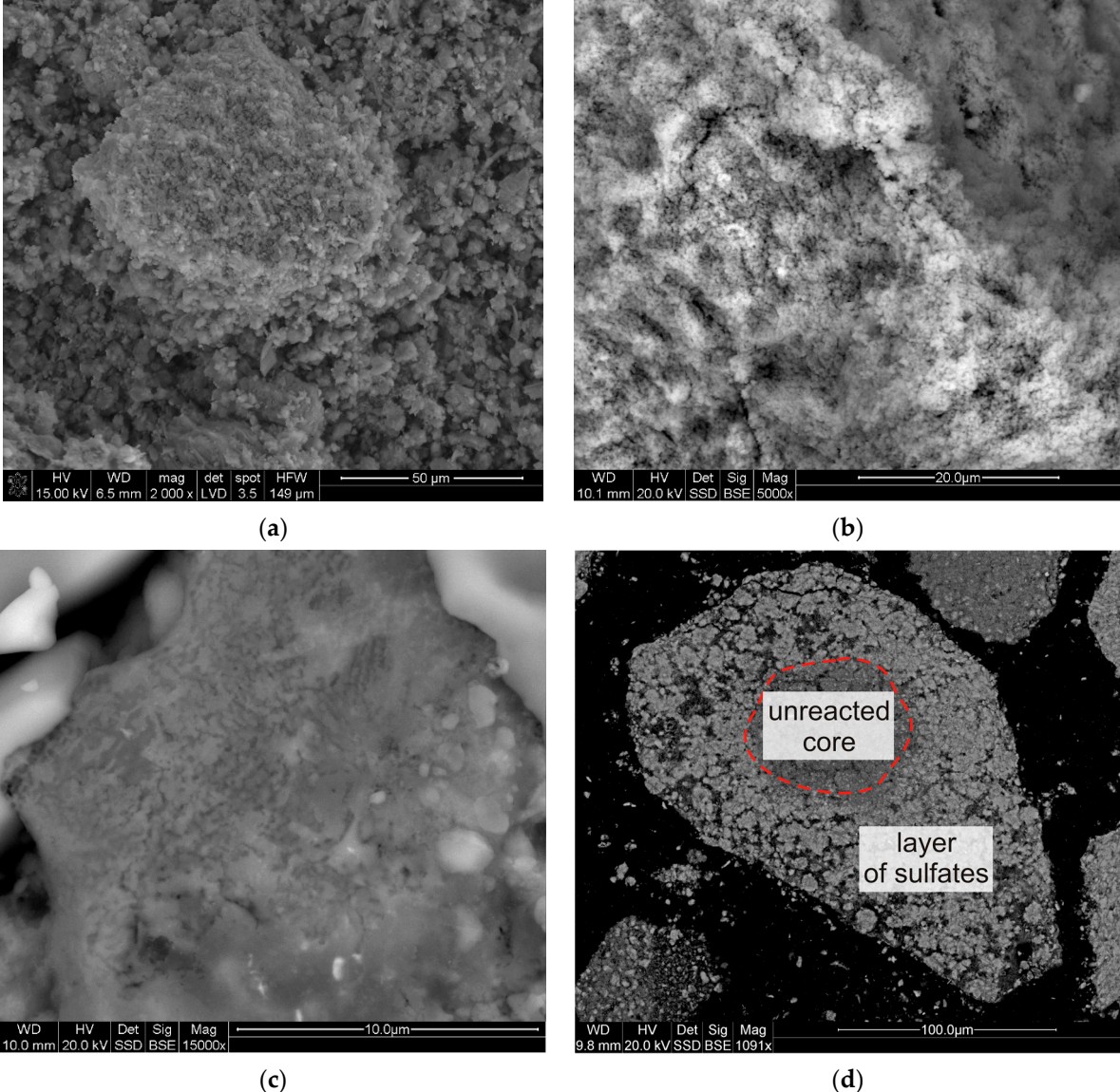

**Figure 11.** Magnesite from Braszowice: (**a**) natural sample; (**b**) samples after the decarbonation process; (**c,d**) after the sulfation process; (**c**) grain surface; (**e**) cross-section through the grain.

To determine the influence of the sorbent texture parameters on the $SO_2$ sorption efficiency, tests of the specific surface area and porosity of magnesites from Wiry and Braszowice were carried out using low-temperature nitrogen sorption and mercury porosimeter. The following were tested: natural samples and samples after the decarbonization and sulfation processes. The test results were compared with an industrial sorbent (Table 7).

**Table 7.** The porous texture parameters determined by low-temperature nitrogen sorption of individual lithological varieties of the magnesites form Wiry and Braszowice and industrial sorbent after the decarbonation process.

| Parameters | | Unit | Magnesite | | Industrial Sorbent |
|---|---|---|---|---|---|
| | | | Wiry | Braszowice | |
| (1) | $S_{BET}$ | | 4.31 | 12.90 | 1.22 |
| | $S_{POR}$ (1) | | 1.22 | 2.28 | 0.24 |
| (2) | $S_{BET}$ | $[m^2/g]$ | 8.12 | 21.97 | 5.33 |
| | $S_{POR}$ (2) | | 4.02 | 5.54 | 3.14 |
| (3) | $S_{BET}$ | | 1.24 | 8.65 | 0.85 |
| | $S_{POR}$ (3) | | 0.79 | 2.67 | 0.52 |
| $V_{BET}$ (2) | $V_{tot}^{0.99}$ | $[cm^3/g]$ | 0.119 | 0.165 | 0.081 |
| | $V_{mik}^{DR}$ | | 0.050 | 0.121 | 0.021 |
| | $V_{mik}^{DR}/V_{tot}^{0.99}$ | - | 0.420 | 0.733 | 0.259 |
| | $V_{mez}^{BJH}$ | $[cm^3/g]$ | 0.056 | 0.032 | 0.055 |
| | $V_{mez}^{BJH}/V_{tot}^{0.99}$ | - | 0.471 | 0.194 | 0.679 |
| | $V_{mak}$ | $[cm^3/g]$ | 0.013 | 0.012 | 0.005 |
| | $V_{mak}/V_{tot}^{0.99}$ | - | 0.109 | 0.073 | 0.062 |
| $P_{POR}$ | (1) | [% vol.] | 42.11 | 16.18 | 43.02 |
| | (2) | | 62.23 | 21.32 | 70.17 |
| | (3) | | 21.22 | 11.11 | 32.80 |

Explanations: the pore size ranges are given according to the pore classification introduced by the International Union of Pure and Applied Chemistry (IUPAC)—micropores: <2 nm; mesopores: 2–50 nm; macropores: >50 nm. BET—low-temperature nitrogen sorption method; POR—mercury porosimeter method; S—specific surface $(m^2/g)$; $V_{tot}^{0.99}$—total pore volume; $V_{mik}^{DR}$—micropore volume. $V_{mik}^{DR}/V_{tot}^{0.99}$—share of micropores in the total pore volume; $V_{mez}^{BJH}$—mesopore volume; $V_{mez}^{BJH}/V_{tot}^{0.99}$—share of mesopores in the total pore volume; $V_{mak}$—macropore volume; $V_{mak}/V_{tot}^{0.99}$—share of macropores in the total pore volume; P—effective porosity; (1)—natural sample; (2)—sample after the decarbonation process; (3)—sample after the sulfation process.

The analysis of the porous texture parameters of the tested samples indicates that the $SO_2$ binding efficiency does not directly depend on the value of the sorbent specific surface area. Magnesite from Braszowice has the largest surface area: $S_{BET}$—12.9 $m^2/g$, $S_{POR}$—2.28 $m^2/g$ before the decarbonation process and $S_{BET}$—21.97 $m^2/g$, $S_{POR}$—5.54 $m^2/g$ after the decarbonization process (Table 6). In this case, the porosity and, above all, the diameters of the pores formed during the sorbent decarbonization process are important. The analysis of the porosity of magnesite from Braszowice in connection with the low sorption efficiency indicates that the presence of micropores (i.e., pores below 2 nm) is not desirable. The specific surface of the magnesite from Braszowice was developed mainly based on micropores ($V_{mik}$—0.121 $cm^3$; $V_{mik}/V_{tot}$—73.3%). As a result, the effective porosity of this magnesite is lower (Table 6). The distribution of pore volume as a function of their diameter in the magnesites from Braszowice, presented in Figure 12, shows that pores with diameters below 0.1 μm, which were formed in the decarbonization process (green line), were not filled in the sulfation process (red line). It should be assumed that they do not participate in the $SO_2$ sorption process. In the case of magnesite from Wiry, the diameters of the pores are above 0.1 μm (Figure 13). The low efficiency of $SO_2$ sorption of magnesite from Braszowice is related to the unfavorable nature of the porous texture, i.e., a high proportion of pores in the range of micropores and smaller mesopores (below 0.1 μm).

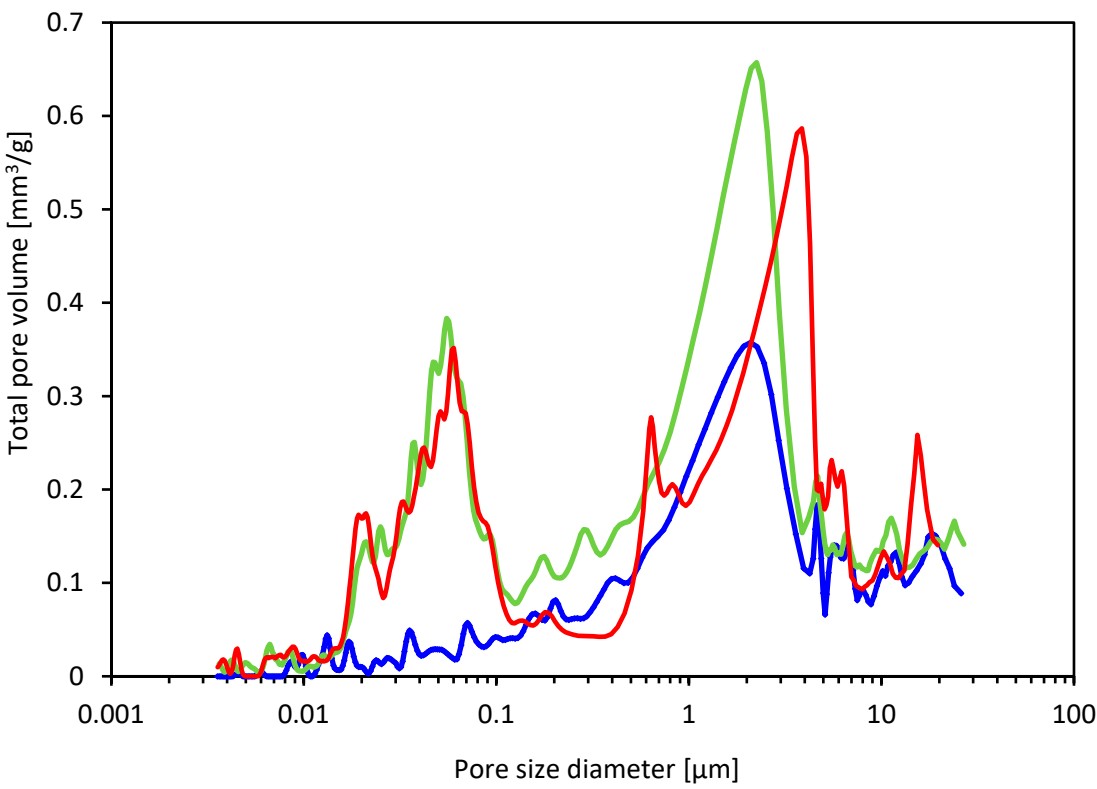

**Figure 12.** Magnesite from Braszowice. Pore volume distributions as a function of their diameter (–natural sample; –after the decarbonation process; –after the sulfation process).

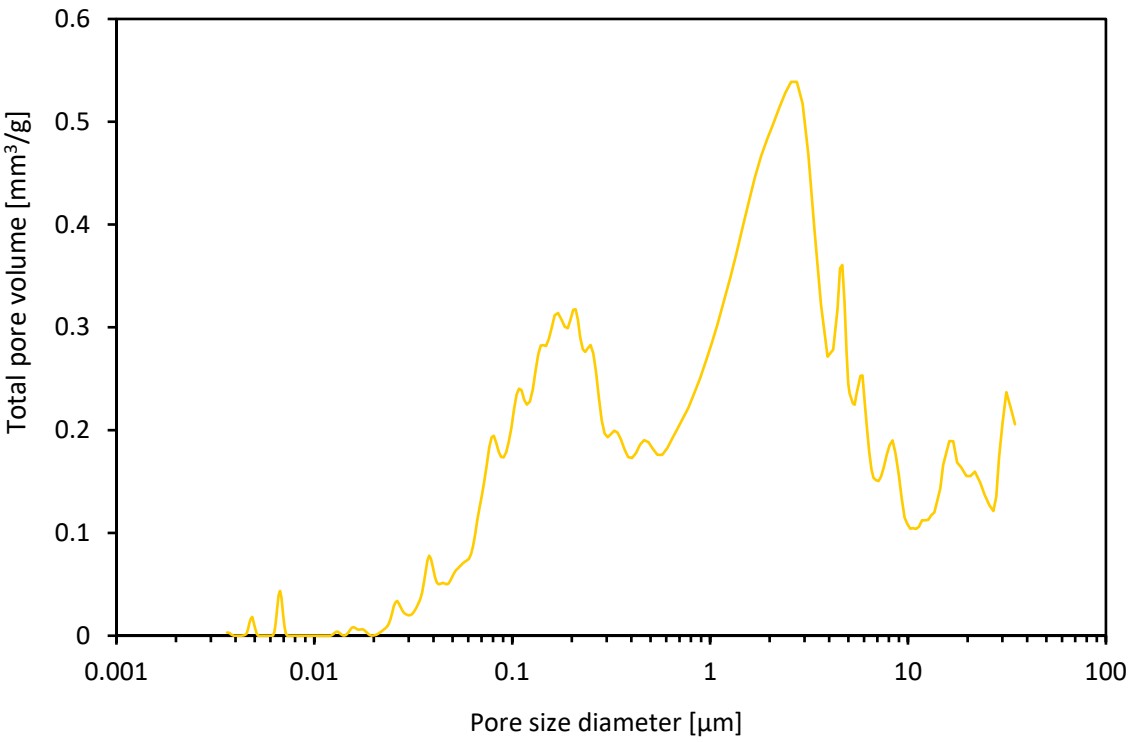

**Figure 13.** Magnesite from Wiry. Pore volume distribution as a function of their diameter after the decarbonation process.

## 4. Conclusions

The magnesites, as well as limestones and dolomites, under conditions of fluidized bed combustion technology can act as $SO_2$ sorbents. The $SO_2$ sorption efficiency of magnesites corresponds to that of limestone and is lower than that of dolomites. However, it should be mentioned that dolomites show the highest sorption efficiency among carbonate rocks [9].

Magnesium as MgO takes an active part in binding $SO_2$ to form $MgSO_4$ and $CaMg_2(SO_4)_3$ sulfate phases. The resulting sulfation products are stable in the temperature conditions characteristic of the fluidized bed combustion technology. The decomposition of $MgSO_4$ begins at a temperature of approx. 950 °C, and the highest rate of this process occurs at approx. 1050 °C. The decomposition of $CaMg_2(SO_4)_3$ is observed at a temperature of approx. 1120 °C.

The high content of carbonate components does not guarantee high desulfurization efficiency. In the $SO_2$ binding process, the textural parameters of the sorbent are important, especially the diameter of the pores formed during the decarbonation process. The presence of pores below 0.1 μm is not desirable. Pores with such diameters do not participate in the $SO_2$ capture process. Their presence contributes to the formation of a massive layer of sulfates on the outer surface of the formation of sorbent grains and is responsible for the low degree of sorbent utilization and the high content of unreacted portions of MgO and CaO in the composition of desulfurization products. In real conditions, where decarbonization and sulfation processes occur almost simultaneously, it is also responsible for the presence of undissociated portions of $MgCO_3$ and $CaCO_3$. The process of $SO_2$ sorption with the use of carbonate sorbents (magnesites, limestones, dolomites) is based on a typical non-catalytic reaction in the gas–solid system [4]. Despite the fact that during the laboratory tests, the sulfation process was applied to previously decarbonated sorbents, and the process itself was carried out in stable conditions (temperature, gas flow rate), the $SO_2$ binding process was not uniform. The conditions changed due to the gradual consumption of the sorbent. The porosity of the sorbent had a decisive influence on the dynamics of such changes. In real conditions of fluidized bed furnaces, where the processes of decarbonization and sulfation run in parallel, more dynamic changes during the sulfation reaction should be expected [29].

The course of the sulfation process in real conditions can be divided into at least three stages. The effectiveness of $SO_2$ binding occurring during individual stages will be shaped by the porosity of the sorbent [30,31]. First, through the primary pores, related to the textural nature of the rock, and then through the secondary ones, formed during decarbonation.

1st stage—simultaneous diffusion of $SO_2$ into the sorbent grains and $CO_2$ from the inside of the sorbent grains (mainly through primary pores related to rock porosity and secondary pores formed in high-temperature conditions because of carbonate decomposition) combined with $SO_2$ adsorption on the outer surface of the sorbent grains;

2nd stage—simultaneous adsorption of $SO_2$ mainly on the external, decarbonated surface of the sorbent grains and on the internal surface of the pores formed during the decarbonation process, combined with the release of $CO_2$ from the inside of the sorbent grains;

3rd stage—diffusion of $SO_2$ inside the sorbent grains through the layers of desulfurization products formed on their outer surface.

In the initial phase of sulfation, the gas–solid reaction process proceeds rapidly. $SO_2$ binding occurs most intensively on the outer surface of the sorbent grains and leads to the formation of a layer of desulfurization products. Over time, $SO_2$ diffusion through the layer of the desulfurization product begins to play a greater role, and the $SO_2$ binding process is transferred to the interior of the sorbent grains [32]. It is slower, because the product of desulfurization both on the outer surface of the grains and on the inner surfaces of the pores formed during decarbonation hinders the flow of $SO_2$ by blocking the sorbent pores [29,33]. If the sorbent secondary porosity is developed with smaller-diameter pores (in the range of micropores and smaller mesopores), $SO_2$ diffusion into the interior of sorbent grains will be stopped. The $SO_2$ binding process will take place only on the outer surface of the

sorbent grains until the free portions of CaO and MgO are reached. In extreme cases, if the share of larger pores is very limited, the diffusion of $SO_2$ into the sorbent grains and $CO_2$ from the inside of the sorbent grains will already be stopped in the initial phase of desulfurization. This will result in the presence of not only unreacted portions of MgO, but also undecomposed $MgCO_3$.

In evaluating the sorption properties of the investigated magnesites, the presence of clay minerals cannot be ignored. Clay minerals should not be treated simply as non-reactive ballast, not involved in $SO_2$ sorption. A significant content of illite and montmorillonite in magnesite from Braszowice unfavorably shapes the texture of the sorbent, which additionally affects the low efficiency of $SO_2$ binding. Clay minerals, by sealing the pore space of the rock, additionally contributed to an increase in the share of micropores and a decrease in the effective porosity value. In addition, they have a negative impact on the decarbonation process. The processes of dehydration and dehydroxylation of illite and montmorillonite occurring during heating caused the effect of pores "shrinking" [34]. This is reflected in the values of the sorbent texture parameters after the decarbonatization process (high proportion of micropores, low value of effective porosity).

**Author Contributions:** Conceptualization. E.H.; Formal analysis. E.H. and M.S.; Investigation. E.H., M.S. and T.R.; Methodology. E.H. and M.S.; Resources. E.H. and M.S.; Supervision. T.R.; Writing—original draft preparation. E.H.; Visualization—E.H. and M.S.; Writing—review & editing. E.H., T.R. and M.S. All authors have read and agreed to the published version of the manuscript.

**Funding:** This research received no external funding.

**Data Availability Statement:** Not applicable.

**Acknowledgments:** This work was carried out in the statutory research of the Department of Mineralogy, Petrography and Geochemistry of AGH in Krakow and the Institute of Mineral and Energy Economy of the Polish Academy of Sciences in Krakow. The research was made with the support of the apparatus of the Center of Energy AGH.

**Conflicts of Interest:** The authors declare no conflict of interest.

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
