# Peer review of "Magnesite as a Sorbent in Fluid Combustion Conditions—Role of Magnesium in SO2 Sorption Process"

_minerals, doi:10.3390/min13030442_

Round 1

Reviewer 1 Report

1. The abstract does not disclose all the studies, so the authors are invited to improve it.

2. The introduction contains important information, but it is too short and does not fully cover the issue under study. The purpose of the work is not formulated. The authors should restructure this part.

3. There are few references to new articles in the work. Of the 27 links in the last 5 years, only 4 links, and in the last 10 years - only 7 links. Authors need to increase the number of links to newer articles.

4. There are some typos throughout the manuscript (for example, in Line 301).

5. There are some grammatical errors and punctuation throughout the manuscript. It is recommended to revise the language.

Author Response

All comments of the Reviewer were taken into account.

1. The new abstract has been inserted.
2. Improved the introduction, added a description of the research objective.
3. Literature supplemented with new items
4-5. Corrected.

Reviewer 2 Report

1.      In the materials and methods section, parts of the writing are the same as shown in an earlier publication by the same authors, i.e., Ref [9]. Examples include the content in Lines 124-133, Lines 143-153, and Lines 180-192. The Reviewer understood that the authors used the same or similar experimental procedure in these two papers. However, it is strongly recommended that the authors should paraphrase these writings to avoid the self-plagiarism problems.

2.      The authors concluded that the presence of clay minerals led to a reduction in the possibility of SO2 adsorption at the end of this paper. More evidence needs to be provided to support this statement. Also, it would be helpful to explain how the clay minerals impede the SO2 adsorption in the results and discussion section.

3.      There are some formatting issues in this paper. For example, the decimal point should be period, not comma. In Figures 7 and 12, the explanation for each color should be added in the figure caption so that readers can readily understand the results. In Figure 13, the titles of x- and y-axis are not in English.

Author Response

All comments of the reviewer were taken into account.

  1. Re-edited Materials and Methods section. It was not possible to completely rule out similarities (formulas, explanations of formulas, definitions).
  2. It explains how clay minerals negatively affect the efficiency of sorption.
  3. Corrected.
